# Therapeutic patient education programs on diabetes in sub-Saharan Africa: A systematic review

**Omomene Iwelomen[1]☯, Jean Toniolo** **[1,2]☯\*, Pierre-Marie Preux[1], Pascale Beloni[1,2]**

**1** Institute of Epidemiology and Tropical Neurology, Inserm U1094, IRD UMR270, University Limoges, CHU Limoges, EpiMaCT - Epidemiology of Chronic Diseases in Tropical Zone, OmegaHealth, Limoges, France, **2** Faculté de Médecine et Pharmacie, Département Universitaire de Sciences Infirmières, Université de Limoges, Limoges, France

☯ These authors contributed equally to this work.
\* jean.toniolo@chu-limoges.fr

**Data Availability Statement:** All relevant data are within the manuscript and its Supporting information files.

## Abstract

### Background

Diabetes is a chronic disease associated with the potential for blindness, kidney failure, heart attacks, strokes, and lower limb amputations. The global prevalence of diabetes is rising, particularly in the sub-Saharan African (SSA) region, where accessing treatment and antidiabetic drugs is complex, leading to challenges in managing the condition. Intentional and structured therapeutic education has demonstrated its ability to enhance health outcomes in diabetes patients. Given the numerous healthcare deficiencies in sub-Saharan Africa, the authors have reevaluated the role of therapeutic patient education (TPE) in this context.

### Methods

This systematic review adhered to the Preferred Reporting Items for Systematic Reviews and Meta-Analysis (PRISMA) guidelines. We queried four databases between March 14 and June 30, 2023 and conducted Cochrane's Risk of Bias analysis on the included studies. Subsequently, a qualitative synthesis of the results was performed.

### Results

The final analysis included thirteen studies. Seven of these, which assessed glycemic control, reported statistically significant results. Additionally, other clinical parameters such as body mass index (BMI), blood pressure, and lipid levels also exhibited some significant improvements. Knowledge substantially increased following the intervention, while attitude, self-care practices, and medication adherence showed no significant improvements. Nurse-led and peer-led intervention programs produced positive outcomes, whereas technology-based intervention methods did not yield favorable results.

**Funding:** The author(s) received no specific funding for this work.

**Competing interests:** The authors have declared that no competing interests exist.

## Conclusion

TPE programs in sub-Saharan Africa have a significant impact on both clinical and non-clinical outcomes in diabetes patients. However, the sustainability of these outcomes remains uncertain. Further research is needed to assess the long-term effects of TPE on diabetes patients.

## Introduction

Diabetes mellitus (DM) is a chronic metabolic disease characterized by high blood sugar level, leading over time to severe damage to the heart, blood vessels, eyes, kidneys, and nerves [1]. Diabetes is one of the world's greatest public health problems, imposing a heavy global burden on public health as well as socio-economic development [2]. The sub-Saharan Africa (SSA) region is expected to experience the largest percentage increase in diabetes incidence of any region in the world [3]. In 2015, the International Diabetes Federation (IDF) estimated that there were 24 million adults with diabetes in the sub-Saharan African region in 2021 [4]. This is a significant increase from the 14.2 million people with diabetes in 2015. The figures are estimated to increase to 33 million by 2030 and 55 million by 2045 [4].

Despite the earlier perception of low non-communicable disease (NCD) mortality rates, current evidence suggests that SSA is now on the cusp of epidemiological transition with a contemporary double burden of disease from NCDs associated with persistent infectious disease [5]. Yet external and internal funding for health care needs continues to focus on communicable diseases while neglecting the needs of large populations suffering from excessive morbidity and mortality from NCDs [6].

The increasing number of people with diabetes in SSA has been attributed to the aging population, dietary and lifestyle changes [7]. Some risk factors for Type 2 diabetes mellitus (T2DM) such as obesity, poor diet, insufficient physical activity, alcohol consumption and smoking are modifiable by behavioral and environmental changes [8]. The main goal of diabetes treatment is to keep blood sugar levels normal and prevent complications [9].

Healthcare providers tend to talk to patients about their illness rather than training them in day-to-day management of their condition, resulting in low self-management knowledge [10]. Therapeutic patient education (TPE) is a relevant approach to this problem while allowing patients to develop skills to better manage their condition. A diabetes TPE program would aim to educate patients on; choice of healthy lifestyles (healthy diet, physical activity, smoking cessation, weight management and effective strategies to cope with stress); Self-management of disease (taking and management of medications and, when clinically appropriate, self-monitoring of blood sugar and blood pressure) as well as prevention of diabetes complications (self-monitoring of foot health; active participation in screening for diseases of the eye, foot and kidney complications) [9]. Studies have shown the impact of TPE as a clinically and cost-effective solution to improve biomedical and psychosocial outcomes in people with metabolic disorders [11–14].

With the many gaps in care delivery in SSA, there has been little effort to build the structures needed to promote safe and effective self-management strategies [15]. This raises questions about the involvement of TPE programs in a context where access to care and quality treatment is limited and expensive and its impact on people with diabetes to self-manage their condition and reduce complications.

A review was conducted to assess the level of self-management in people with diabetes in sub-Saharan Africa [16]. Another study exposed barriers to patient nonadherence to self-

management intervention recommendations in Africa [17]. A recent scoping review examined diabetes self-management education interventions in the WHO African Region, offering valuable insights [18]. While similar, our study employs a more systematic methodology and aims to provide a unique perspective, enhancing and expanding existing literature to deepen our understanding of the challenges and opportunities specific to the SSA region. Our review serves to provide an overview of diabetes TPE programs in SSA, exposing actors, methods, and relative outcomes on patient health status. The question in this review was: what role do TPE programs play in health-related outcomes of persons with diabetes mellitus in SSA compared to standard care?

## Materials and methods

### Search strategy

This systematic review was conducted following the Cochrane guidelines for conducting systematic reviews [19] and is reported using the PRISMA guidelines for writing and reading literature reviews and meta-analyses [20]. This review was registered on PROSPERO (ID: CRD42023440701). Four electronic databases (PubMed, Web of Science, CINAHL and Google Scholar) were systematically searched for relevant publications between March 14 and June 30, 2023. A preliminary search was performed to derive keywords based on the PICO search question then 'title and abstract search' was performed on each database and Boolean operators were used when applicable.

The keywords for each database queried were composed of: "patient education", "self-management education", "educational therapy", "therapeutic education", "therapeutic patient education", "patient compliance", "self-management", "diabetes self-care", "intervention program", "diabetes", "diabetes mellitus", "sub-Saharan Africa", "SSA", "sub-Sahara".

For PubMed, MeSH terms and the name of each country in SSA following the Library of Congress list [21] were included. The other databases were searched with the same keywords, but the search strategy was adapted to the specificities of each database. Databases were searched and articles were selected independently by two researchers under the supervision of a third experienced researcher. A table summarizing the different search strategy is available in S1 Appendix.

### Screening and selection criteria

After the database search, the results were exported to Zotero, a bibliographic management software, where duplicates were removed and then moved to Rayyan QCRI software for title and abstract screening. Studies were independently reviewed and assessed for eligibility by two authors (OI and JT). A third researcher was consulted to resolve any disagreements that arose (PB). The final step was to review eligibility by reading the full-text articles to determine which would be included and which would not. Studies were included in this review if they met the following inclusion criteria; 1) They took place in at least one country in sub-Saharan Africa, 2) Participants were people with type 1 or type 2 DM, 3) The study evaluated a TPE intervention versus standard care, 4) The study measured change in a health-related outcome.

All published intervention studies—RCTs, non-RCTs, quasi-experimental, were included if they met the criteria. There were no limitations on the date and language of the article published or the impact factor of the journal. Qualitative studies, observational studies, uncontrolled studies, systematic reviews, and grey literature were not included.

## Data extraction and analysis

Relevant data were extracted from eligible articles using a Microsoft Excel spreadsheet. These included author name, date of publication, journal, and country of study; the main objective of the study; study design and participants; key outcome measures; composition, method, duration, and frequency of intervention; main results and limitations cited in the studies.

A risk of bias analysis was performed on each of the included studies using the Cochrane 'Risk of Bias' 2.0 tool [22]. A risk of bias assessment was performed in 5 domains (randomization process, blinding, missing outcome data, outcome measurement and outcome reporting). The risk of bias judgments for each domain are 'low risk of bias', 'some concerns' or 'high risk of bias'. Judgments are based on responses to signaling questions. We adopted the risk of bias tool to assess non-RCTs included in the review to expose biases inherent in the study design.

A qualitative synthesis was performed after data extraction and quality assessment.

Ethical approval was not required for this study.

## Results

### Description of selected studies

Out of a pool of 2,497 articles identified from the database, 25 full-text articles were assessed for eligibility. Four articles were excluded due to mismatches in the study population—not in SSA region (n = 1) and study design (qualitative, cohort, no control group and non-related interventions) (n = 8) (Fig 1). The final analysis included 13 distinct studies from 16 articles, all published between 2010 and 2021. These studies encompassed various research designs: 11 randomized controlled trials, 1 quasi-experimental study, and 1 non-randomized controlled trial. The sample sizes of these studies ranged from 28 to 1,569 patients.

In total, 3,017 persons with diabetes participated in these studies, with 932 being male and 2,085 female participants. Twelve studies exclusively recruited adult participants, while one study focused on children and young adults. Nine studies centered on Type 2 Diabetes Mellitus (T2DM), one focused solely on Type 1 Diabetes Mellitus (T1DM), and three included both persons with T1DM and persons with T2DM. Seven studies were conducted in a tertiary hospital, 2 in secondary institutions and 4 in a primary health care facility. The geographical distribution of these studies was as follows: South Africa (n = 5), Nigeria (n = 2), Cameroon (n = 1), Ethiopia (n = 1), Rwanda (n = 1), Mali (n = 1), Kenya (n = 1), and Ghana (n = 1). Table 1 provides an overview of the characteristics of each study, along with pertinent results.

### Composition and methods of intervention

The education contents of the interventions were tailored or corresponded with the American Association of Diabetes Educators 7 core themes in diabetes education: Health eating, being active, self- monitoring, medication taking, problem solving, reducing complications and stress management [40]. However, two interventions were focused on nutrition and one study was a physical activity and dietary intervention [24,27,28,39]. Although multiple themes were incorporated into each intervention, all studies shared a central objective: evaluating the impact of patient education programs on the health outcomes of persons with diabetes. The primary outcome assessed in all studies was the clinical health status of participants. Some studies also evaluated participants' knowledge, [24,28,31,33,35,37,39] attitudes, and adherence to recommendations [25,28,31,37,38] reduction of acute complications [28,35], and the feasibility and acceptability of technology-based intervention methods [35,36,38].

Various theories, models, and frameworks guided the development of educational programs in these studies. These included self-determination theory, cognitive theory, and

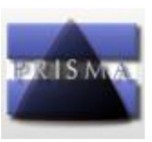

# PRISMA 2009 Flow Diagram

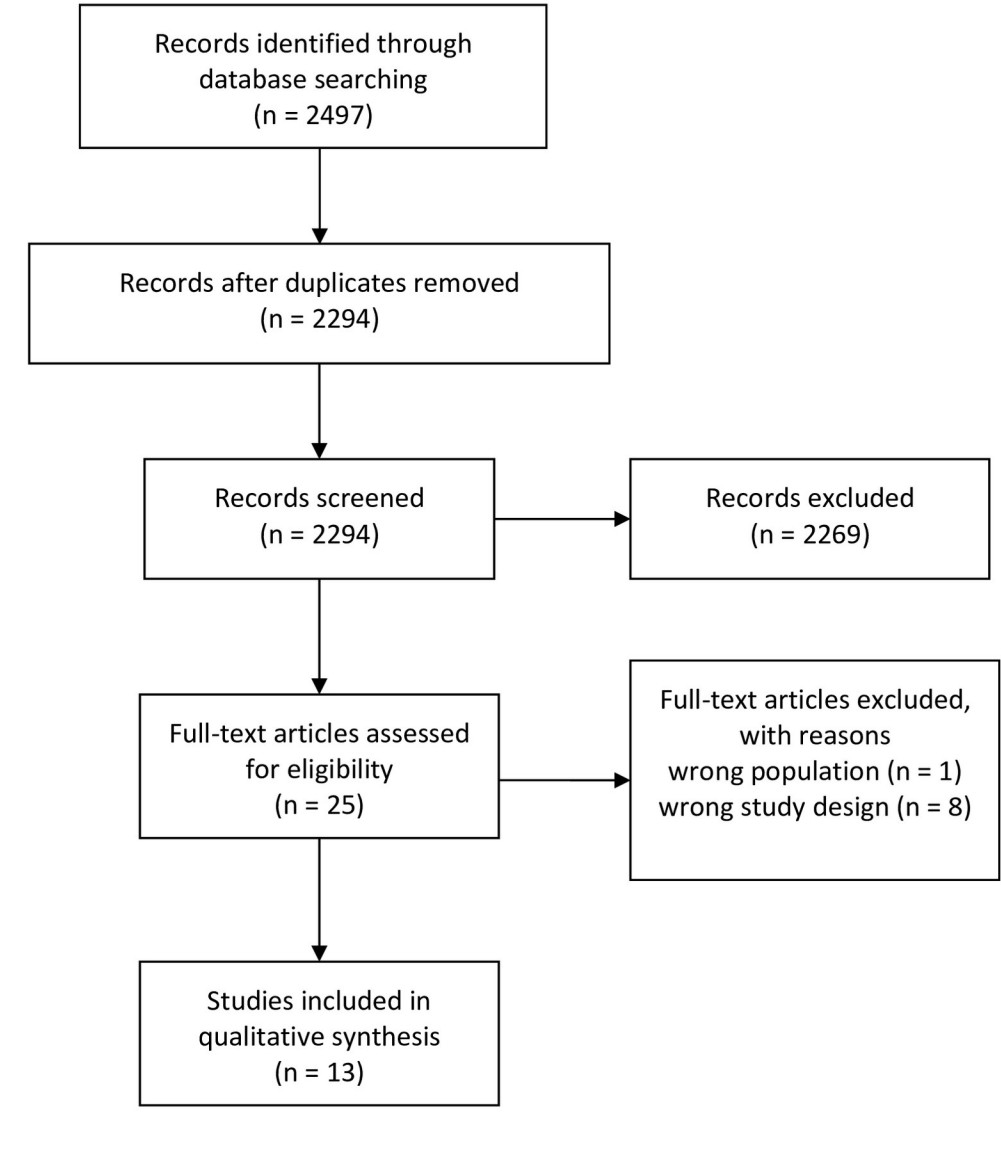

**Fig 1. PRISMA flowchart.** Moher et al.[23].

**Table 1. Description of the characteristics of the studies in the systematic review [24–39].**

| ID | First author (date) Journal Location | Objective | Study design Population | Outcome measure Assessment period | Intervention: C: Composition M: Method I: Instructor D: Duration F: Frequency (total) | Result | Limits |
|---|---|---|---|---|---|---|---|
| 1 | AJ et al. (2010) *South African Journal of Physiotherapy* South Africa[24] | Establish the effectiveness of a cost-effective daily walk and diet education intervention programme | RCT-NB *n = 43 adults 17 M 26 F T2DM* | Mean difference in HbA1c, total cholesterol and LDL cholesterol. Mean difference in diabetes knowledge *Baseline, 16 weeks and 1 year* | C: Dietary and PA education, motivation text messages M: Group session I: Physiotherapist, podiatrist, and dietician D: 4 weeks F: Weekly (4) | ↘ HbA1c at 16 weeks significant (p = 0.041)↔ HbA1c at 1 year ↘ Total cholesterol and LDL-cholesterol at 1 year significant (p = 0.047; p = 0.014) ↑Knowledge score significant at 16 weeks and 1 year | High attrition rate |
| 2 | Mash et al. (2012) *BMC Family ¨Practice* South Africa[25] | Evaluate effectiveness of group diabetes education on diabetic health outcomes | RCT-NB n = 1570 adults 411 M 1158 F T2DM | 1% reduction in HbA1c 5% weight loss mean difference in BP and self-care activities *Baseline and 12 months* | C: General diabetes information, lifestyle education, medication use and complication management M: Group session I: Health promotion officer D: 6 months F: Monthly (4) | ↔ HbA1c, weight loss, self-care activities ↘ SBP and DBP significant (p = 0.04; P = 0.002) | High attrition rate Low attendance rate |
| 3 | Afemikhe et al (2015) *Africa Journal of Nursing & Midwifery* Nigeria[26] | Determine the effects of a structured multidisciplinary patient centred DSME programme | Quasi-experimental study *n = 28 adults 11 M 17 F T2DM* | Mean difference in FBS levels and BMI *Baseline and 6 weeks* | C: PA and NE, SBGM instruction and lifestyle education M: Group and individual sessions I: Nurses, dieticians D: 5 weeks F: Weekly (5) | ↘ FBS levels significant p = 0.012 ↔ BMI | Small sample size High attrition Short follow-up period |
| 4 | Muchiri et al. (2016) *Public Health Nutrition* South Africa[27] | Evaluate a nutrition programme's effect on HbA1c, BMI, blood pressure, blood lipids and dietary behaviour | RCT-NB n = 82 adults 11 M 71 F T2DM | Mean difference in HbA1c, blood lipid, BP, BMI, and dietary behaviour *Baseline, 6 months, 12 months* | C: NE M: Group sessions and follow-up I: Dietician, Nutrition science student and Horticulture officer D: 8 weeks F: Weekly (8) | ↔ HBA1c, BMI, lipid profile, BP ↘ Starchy-food intake significant at 6 and 12 months (p = 0·005; p = 0·017) ↘ Median energy intake significant (p = 0·017) ↔ Other dietary behaviour | Small sample size Non fasting blood samples non-expert instructors |
| | Muchiri et al. (2016) *Journal of Endocrinology Metabolism and Diabetes of South Africa* South Africa[28] | Evaluate the effect of a nutrition education programme on diabetes knowledge and attitude | RCT-NB n = 82 adults 11 M 71 F T2DM | Significant change in knowledge and attitude towards diabetes and treatment *Baseline, 6 months, and 12 months* | C: NE and general diabetes knowledge education M: Group sessions and follow-up I: Dietician, Nutrition science student and Horticulture officer D: 8 weeks F: Weekly (8) | ↑ Mean diabetes knowledge scores significant at 6 and 12 months (p = 0.033; p < 0.001) Patient autonomy, the only score significantly higher at 12 months (p = 0.028) in patient attitude | Non validated questionnaire |

*(Continued)*

**Table 1.** (Continued)

| ID | First author (date) Journal Location | Objective | Study design Population | Outcome measure Assessment period | Intervention: C: Composition M: Method I: Instructor D: Duration F: Frequency (total) | Result | Limits |
|---|---|---|---|---|---|---|---|
| 5 | Etienne et al. (2017) *Diabetes Research and Clinical Practice* Rwanda [29] | Assess the efficacy of a lifestyle education programme compared to the current tertiary standard of diabetic care | RCT-NB n = 251 adults77 M 174 F T1DM, T2DM | Mean difference in HbA1c, BMI, FBS, SBP, DBP *Baseline and 12 months* | C: Education on adapted diet, PA; lifestyle, SBGM and medication adherence M: Group session I: Physicians, nurses, nutritionists, and psychologists D: NR F: Monthly (minimum 8) | ↘ HbA1c statistically significant (p<0.001) ↘ FBS, SBP, DBP, BMI significant at 12 months (p <0.001, 0.005, 0.02, <0.001 respectively) | Possible case mix |
| 6 | Essien et al. (2017) *Plos One* Nigeria[30] | Determine the effectiveness of a structured, guideline based DSME intervention on clinical outcomes | RCT-NB n = 118 adults 47 M 171 F T1DM, T2DM | Mean difference in HbA1c *Baseline and 6 months* | C: Instructions on nutrition, medication compliance, exercise foot care and lifestyle behaviour change M: Group session I: Doctors, nurses D: Six months F: Fortnightly (12) | ↘↘HbA1c between intervention and control group significant (p < 0.0001) | Short follow-up period |
| 7 | Debussche et al (2018) *Plos One* Mali[31] | Evaluate the effectiveness of peer-led self-management education in improving glycaemic control | RCT-NB n = 141 adults 36 M 115 F T2DM | Mean difference in HbA1c, BMI, SBP, DBP Improvement in dietary knowledge, SBGM and treatment *Baseline, 3, 6 and 12 months* | C: Dietary and PA education, complication management M: Group session I: Trained peer educators D: 1 year F: Quarterly (4) | ↘ HbA1c levels, BMI, SBP significant at 12 months (p = 0.006; p = 0.0005; p = 0.003) ↔ Diet and knowledge scores | Possible cross-contamination Urban setting; not generalizable |
| 8 | Hailu et al. (2018) *Frontiers in Public Health* Ethiopia[32] | Determine the effects of DSME on clinical outcomes | RCT-NB n = 220 adults 148 M 72 F T2DM | Mean difference in proportion of participants with HbA1c ≤ 7% Significant change in FBS, SBP, DBP *Baseline and 9 months* | C: general knowledge, nutrition and PA education, stress management M: Group session I: Nurses D: 6 months F: Monthly (6) | Mean difference in proportion of participants with HbA1c not significant ↔ FBS, SBP, DBP levels | Low attendance rate Possible selection bias and information spill-over |
| | Hailu et al. (2019) *Diabetes Metabolic Syndrome and Obesity-Targets and Therapy* Ethiopia[33] | Test the effectiveness of a multifaceted, nurse led DSME programme for improving diabetes knowledge, self-care activities, and self-efficacy | RCT-NB n = 220 adults 148 M 72 F T2DM | Significant improvement in knowledge, self-care behaviour and self-efficacy *Baseline and 9 months* | C: Dietary recommendation, foot-care, SBGM, lifestyle behaviour and PA education M: Interactive group sessions I: Nurses D: Six months F: Monthly (6) | ↑Diabetes knowledge scores significant (p = 0.044) Adherence to dietary recommendations and performed foot-care significant (p = 0.019; p = 0.009) ↔ Other self-care behaviour and self-efficacy | Short follow-up period Low attendance rate High attrition rate |
| 9 | Gathu et al. (2018) *African journal of primary health care & family medicine* Kenya[34] | Evaluate whether a structured DSME in addition to usual care improves glycaemic control among sub-optimally controlled T2DM patients | RCT-NB n = 140 adults 78 M 62 F T2DM | Mean difference in HbA1c, BMI, BP *Baseline and 6 months* | C: Instruction on self-care behaviour, nutrition, medication adherence, SBGM and foot- care M: Group session I: Certified diabetes educator D: 18 weeks F: Monthly (3) | ↔HbA1c, BMI, BP at 6 months | Short follow-up period Possible cross contamination |

(*Continued*)

**Table 1.** (Continued)

| ID | First author (date) Journal Location | Objective | Study design Population | Outcome measure Assessment period | Intervention: C: Composition M: Method I: Instructor D: Duration F: Frequency (total) | Result | Limits |
|---|---|---|---|---|---|---|---|
| 10 | Sap et al. (2019) *Pediatric diabetes* Cameroon[35] | Evaluate knowledge, glycaemic control, and frequency of acute complications after DSME through a social network of adolescents with diabetes | Non-RCT-NB n = 54 children, adults 29 M 25 F T1DM | Significant improvement in glycaemic control, knowledge and reduction in acute complications *Baseline and 3 months* | C: Insulin and medication use, SBGM, dietary education and complication management M: WhatsApp group sessions I: Medical team D: 4 weeks F: Weekly (4) | ↔ HbA1c levels at 3 months ↑ Knowledge score significant in intervention group at 2 months (p<0.01) Slight decrease in acute complication reported in the intervention group | Small sample size Short follow-up period Self-reported acute complications |
| 11 | Owolabi et al. (2019) *Plos One* South Africa[36] | Determine the efficacy, acceptability, and feasibility of text messaging in promoting glycaemic control and clinical outcome measures | RCT-NB n = 216 adults 34 M 182 F T1DM, T2DM | Mean difference in RBG, BMI, BP Patient satisfactory rate of SMS intervention *Baseline and 6 months* | C: Educative and motivating texts on diabetes and dietary recommendations M: Text messaging I: Research team D: Six months F: Daily | ↔RBG, BMI, BP at 6 months90.74% satisfactory rate in intervention method | RBG as a measure of glycaemic status Result not generalizable to T1DM patients |
| | Owolabi et al. (2020) *Medicine* South Africa[37] | Determine the effect of a unidirectional text messaging on adherence to recommended diets and activity among DM patients | RCT-NB n = 216 adults 34 M 182 F T1DM, T2DM | Difference in medication, dietary and physical activity adherence *Baseline and 6 months* | C: Advice on lifestyle behaviour; diet, PA, medication use and smoking cessation M: Text messaging I: Research team D: 6 months F: Daily | ↔ Mean medication, dietary and PA adherence at 6 months | Self-reported assessment measure; possible response bias |
| 12 | Asante et al. (2020) *The Diabetes educator* Ghana[38] | Evaluate the feasibility and effectiveness of a nurse-led mobile phone call intervention on glycaemic management and adherence to self-management practices | Pilot RCT-NB n = 60 adults 13 M 47 F T2DM | Mean difference in HbA1c Significant improvement in medication adherence and self-management measures *Baseline and 12 weeks* | C: Reinforcement on diet, exercise, SBGM, foot-care and medication adherence M: Telephone calls I: Nurses D: 12 weeks F: Weekly (16) | ↘ Mean HbA1c levels at 12 weeks significant for intervention group (p = 0.004) Mean difference in HbA1c between groups not significant Increase in foot care practice shown but not statistically significant ↔ Other measured outcomes | Short follow-up period Possible selection bias; insulin-taking patients excluded |
| 13 | Muchiri et al. (2021) *Journal of Diabetes and Metabolic Disorders* South Africa[39] | Determine the effectiveness of an adapted nutrition education programme on clinical status and dietary behaviours, on adults with poorly controlled diabetes | RCT-NB n = 77 adults 21 M 56 F T2DM | 0.5% reduction in HbA1c level Significant improvement in BMI, BP, blood lipid and dietary behaviour *Baseline, 6 and 12 months* | C: Adapted NE, general diabetes knowledge education M: Group interactive session, individual session, and follow-up I: Dietician D: 1 year F: Monthly (8) | ↘ HbA1c levels at 6 months clinically significant (0.53%), albeit not sustained at 12 months ↘ SBP, DBP at 12 months and energy intake at 6 months significant (p = 0.004; p = 0.016; p = 0.024) ↑ Diabetes knowledge scores at six months | Small sample size |

↑ Increase; ↘ reduction; ↔ No significant difference; DM, Diabetes mellitus; T1DM, Type 1 diabetes mellitus; T2DM, Type 2 diabetes mellitus; DSME, Diabetes self-management education; NE, Nutrition education; PA, Physical Activity; F, Female; M, Male; SBGM, Self-blood glucose monitoring; HbA1c, Glycated hemoglobin; RCT, Randomized controlled trial; NB, Non-blinded; FBS, Fasting blood sugar; RBG, Random blood glucose; LDL, Low density cholesterol; HDL, High density cholesterol; BP, Blood pressure; SDP, Systolic blood pressure; DBP, Diastolic blood pressure; BMI, Body-mass index; NR, Not reported.

motivational interviewing, as well as the learning nest approach and skilled helper model [24,25,27,31]. Some interventions adapted international guidelines and programs for the provision of TPE to suit local and cultural contexts, particularly regarding nutrition [27,31,39]. Educational materials such as leaflets, brochures, and illustrations facilitated group discussions, which took place in different formats including group sessions (n = 14), individual sessions (n = 4), WhatsApp groups (n = 1), text messaging (n = 4), and phone calls (n = 1). Several programs included follow-up sessions and SMS reminders. The frequency of group sessions varied from 4 to 16 (mean frequency = 8), and the duration of interventions ranged from 4 weeks to 1 year (mean duration = 9 months). Lead instructors for these interventions were nurses (n = 4), dietitians and nutrition specialists (n = 2), health promoters and educators (n = 1), peer educators (n = 1), and multidisciplinary medical teams (n = 5).

## Clinical results and method of evaluation

The clinical outcomes assessed across the studies included glycemic control (n = 13), body mass index (BMI) (n = 11), blood pressure (n = 9), and lipid profile (n = 4). In all studies the outcomes were evaluates by comparing the mean differences between the endpoints of the two intervention arms to baseline. Among the 13 studies measuring glycemic control, six reported a significant mean difference in HbA1c levels [24,29–31,38,39], while one reported a significant mean difference using fasting blood sugar (FBS) [26]. However, no significant difference was found in the study that measured glycemic status using random blood glucose (RBG) [36]. Two out of 11 studies observed a significant reduction in mean BMI in the intervention group compared to the control group [29,31], and five studies reported a significant reduction in blood pressure compared to the control group [25,29,31,32,39]. Only one out of four studies recorded positive results in terms of lipid levels [24].

## Knowledge, attitude, and compliance

Six studies evaluated participants' knowledge of diabetes and self-management post-intervention, obtaining knowledge scores through questionnaires and scales. All but one of the studies reported a significant improvement in participants' knowledge following the intervention or when compared to a control group [31]. Other outcomes assessed included physical activity (n = 4), diet (n = 4), foot care practices (n = 2), and medication adherence (n = 3). Of the four studies measuring physical activity, one observed an improvement in the intervention group [25]. One out of four studies reported positive results regarding dietary patterns and adherence to dietary recommendations [25]. Similarly, one study reported improved foot care practices [38] while no significant positive change was observed in medication adherence [31,37,38].

## Risk of bias and methodological limitations

We conducted a quality assessment of each study using Cochrane's risk of bias tool (Fig 2). In summary, three studies exhibited questionable randomization processes, while two did not perform any randomization. Due to the nature of the interventions, it was not feasible to blind participants and administrators; however, some studies reported blinding of outcome assessors or control groups. Eight out of the 13 studies included in this review were assessed as having a low risk of bias.

Four studies were described as having a small sample size, [26,27,35,39] and four reported high attrition rates [24–26,33]. Two studies that utilized random blood glucose (RBG) and fasting blood sugar (FBS) instead of HbA1c for measuring glycemic status faced limitations in assessing this parameter [26,36]. In two studies, self-reported data was collected to measure

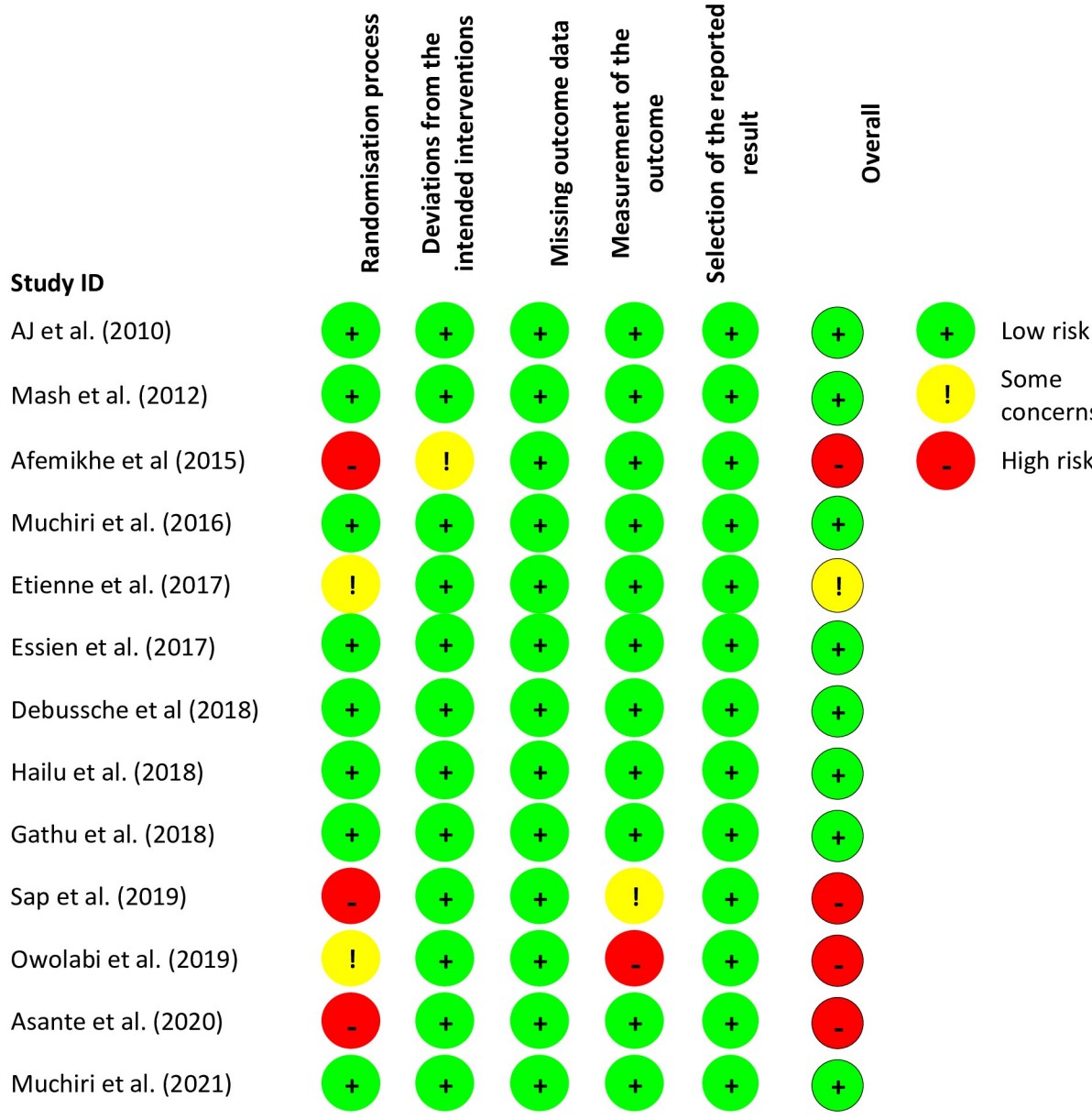

**Fig 2. Risk of bias assessment** [22].

adherence to medication, diet, and physical activity [35,37], while one study employed an unvalidated questionnaire [28].

## Discussion

TPE plays a vital role in managing chronic diseases, such as diabetes [10]. Structured programs in various studies have demonstrated their ability to enhance patient self-management skills, effectively controlling their condition and reducing complications [41,42]. Sub-Saharan Africa (SSA) is on the brink of an epidemiological transition, where non-communicable diseases have become predominant, further adding to the already substantial burden of communicable

disease [5]. As this burden increases, more efficient and effective interventions become imperative. Our study is particularly significant as it offers a comprehensive overview of all structured intervention programs implemented in the Sub-Saharan African region.

Our study comprised 13 experimental controlled studies that assessed structured TPE programs in comparison to control groups receiving standard care. Nearly all the selected studies reported significant results in either clinical or non-clinical parameters. This reinforces positive findings from research conducted in various other global regions and among diverse populations [43–45]. Six studies revealed a noteworthy reduction in HbA1C levels when compared to the control groups [24,29–31,38,39]. Enhanced glycemic control has been shown to relatively reduce cardiovascular complications, as a 1% reduction in HbA1C levels, according to a UK study, can reduce cardiovascular complications by 25% [46]. Moreover, most of the studies addressing the secondary outcome of blood pressure yielded favorable outcomes, which is particularly relevant given that hypertension is a crucial risk factor for diabetic complications. However, substantial results concerning significant weight loss and improved lipid profiles were limited. Many studies reportedly lacked sufficient statistical power to detect significant differences in secondary clinical outcomes [26,27,35,39].

The behavioral aspect is another equally critical facet of evaluating patient therapeutic education. However, the behavioral component as an outcome of the intervention received limited evaluation in most studies. Positive behavioral changes are pivotal for long-term self-management [47]. While the knowledge component was assessed in most studies, with all but one showing significant improvements in patient knowledge through questionnaires and knowledge scales after the intervention, some studies reported that the knowledge level, while significantly better than that of the control group, still fell short [27,39]. This may be attributed to the literacy levels of participants or the methods of instruction. The low baseline knowledge of participants underscores the existing state of diabetes management in the SSA region. Improved knowledge has been directly correlated with enhanced self-care practices [48,49]. However, from the studies, it is evident that improved knowledge did not consistently translate into better adherence. Other factors affecting adherence include socioeconomic status, patient motivation, cost, and time constraints, as indicated by a study on the barriers faced by patients in self-management [17]. These are factors that must be taken into consideration when delivering patient therapeutic education. Some studies reportedly distributed glucometers, pedometers, and logbooks to participants, which were reported to enhance compliance among participants and yield positive clinical outcomes [24–29].

The method, duration, frequency, and facilitators are components of TPE programs that contribute to their success. A combination of group sessions and individual sessions has been found to yield the best results in a previously conducted study for better reinforcement [50]. Various modes of delivery were employed for the interventions in our studies, with group sessions being the most prevalent. Technology-mediated interventions did not yield the expected results; however, only a small number of studies employed this approach [35,36,38]. While multidisciplinary teams are reported to be the most effective facilitators of TPE programs [51], nurse and peer-led interventions have proven effective in various interventions from our study [30,31,32,38]. This is especially important in this region, which faces a shortage of healthcare workers [52]. In situations where doctors are unavailable, nurses can facilitate the programs, and in the absence of nurses, trained peer educators can be a viable option. Previous studies on nurse-led and peer-led interventions in diabetes self-management have yielded positive results [53,54]. A few studies have shown a positive correlation between the program's duration and outcomes [55]; however, our study did not reveal any strong relationship between the frequency or duration of programs and positive outcomes.

The interventions in this study were primarily conducted in tertiary hospitals located in large urban areas and capital cities. This affects the generalizability of the findings to the predominantly rural population that still lacks access to basic care and treatment and has lower literacy levels [56,57]. The participants' literacy level was a recurring exclusion criterion for the interventions. TPE programs should not only be introduced at the primary care level but should also be adapted to reach populations with varying socioeconomic and literacy levels. This can be achieved through communication in native dialects and the use of teaching materials tailored to the cultural context [58]. Diabetes self-care practices predominantly occur within households and can influence diabetes management behaviors [59]. Consequently, there is a need to involve families in these programs [60] In sub-Saharan Africa, one must not disregard the familial context, which remains a source of patient support and motivation [61].

Several limitations were reported in the studies. A significant number of studies reported short follow-up periods. For instance, one study showed significant positive results at four months post-intervention but failed to sustain these results at 12 months [24]. While the primary aim of the program may be to improve clinical outcomes, it is equally important to measure the program's sustainability. Therefore, more studies with longer follow-up periods are required to assess the lasting impact of the intervention on patient self-management competence. In addition, high attrition and low attendance rates of participants also diminished the statistical power of several studies. This is a significant public health concern, along with low referral rates as cited in a study [62]. Qualitative studies should be conducted to comprehend the experiences of patients and their reasons for not participating in these programs, leading to a needs assessment aimed at enhancing program quality and better adaptation to patients' resources.

One of our study's limitations encompasses the heterogeneous nature of the study designs. Non-randomization and the absence of blinding of participants increase the risk of bias in our selected studies. Additionally, there was substantial variability in the educational content and themes of the intervention. Consequently, we did not categorize or synthesize the data collected in the studies based on the theme of each intervention. Each program had a theme that ranged from nutrition education to physical activity and self-blood glucose monitoring, and a combination of themes and activities of varying depth and intensity. Therefore, we were unable to assess which area had a greater or lesser impact. Nevertheless, despite differences in content and delivery mode, most of the interventions reported significant results in one or more clinical parameters. We recognize that nearly all our included studies focused on T2DM, therefore, our synthesis may not be applicable to T1DM patients. It is also worthy to note that our study relied on common health research databases, but a limitation is acknowledged for not including specific African databases like the African Index Medicus. This omission may exclude relevant studies from local journals or not indexed in widely recognized databases. Future research should consider incorporating these databases for a more comprehensive literature exploration.

## Conclusion

TPE programs in sub-Saharan Africa have significant impact on the clinical outcomes of people with diabetes and on improving knowledge and developing skills to self-manage their condition. However, the lasting impact of these programs remains uncertain. Further studies are needed to measure the long-term impacts of TPE programs on patient health status. Extra efforts should be made to include population of all literacy levels and family participation should be built into the programs. The authors also recommend that TPE programs be

implemented at the primary health care level where diabetes in the patient is precocious for the prevention of complications.

**Reporting Method**: PRISMA

## Supporting information

**S1 Appendix. Search strategy.**
(DOCX)

**S2 Appendix. PRISMA checklist.**
(DOCX)

## Acknowledgments

We thank Dr Emilie Auditeau for her help with the organization of this project.

## Author Contributions

**Conceptualization:** Omomene Iwelomen, Jean Toniolo, Pascale Beloni.

**Data curation:** Omomene Iwelomen, Jean Toniolo, Pascale Beloni.

**Formal analysis:** Omomene Iwelomen, Jean Toniolo.

**Investigation:** Omomene Iwelomen, Jean Toniolo, Pascale Beloni.

**Methodology:** Omomene Iwelomen, Jean Toniolo, Pascale Beloni.

**Resources:** Pierre-Marie Preux.

**Software:** Jean Toniolo.

**Supervision:** Omomene Iwelomen, Jean Toniolo, Pierre-Marie Preux, Pascale Beloni.

**Validation:** Omomene Iwelomen, Jean Toniolo, Pierre-Marie Preux, Pascale Beloni.

**Visualization:** Omomene Iwelomen, Jean Toniolo, Pierre-Marie Preux, Pascale Beloni.

**Writing – original draft:** Omomene Iwelomen, Jean Toniolo, Pierre-Marie Preux, Pascale Beloni.

**Writing – review & editing:** Omomene Iwelomen, Jean Toniolo, Pierre-Marie Preux, Pascale Beloni.

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
