## [Decision Letter · Decision Letter 0]

15 Nov 2023

PONE-D-23-34477Therapeutic patient education programs on diabetes in sub-Saharan Africa: A systematic reviewPLOS ONE

Dear Dr. Toniolo,

Thank you for submitting your manuscript to PLOS ONE. After careful consideration, we feel that it has merit but does not fully meet PLOS ONE’s publication criteria as it currently stands. Therefore, we invite you to submit a revised version of the manuscript that addresses the points raised during the review process.

We look forward to receiving your revised manuscript.

Kind regards,

Francis Xavier Kasujja

Academic Editor

PLOS ONE

Journal Requirements:

2. We note that this manuscript is a systematic review or meta-analysis; our author guidelines therefore require that you use PRISMA guidance to help improve reporting quality of this type of study. Please upload copies of the completed PRISMA checklist as Supporting Information with a file name "PRISMA checklist".

Reviewers' comments:

Reviewer's Responses to Questions

**Comments to the Author**

1. Is the manuscript technically sound, and do the data support the conclusions?

Reviewer #1: Yes

Reviewer #2: Yes

2. Has the statistical analysis been performed appropriately and rigorously? 

Reviewer #1: N/A

Reviewer #2: N/A

3. Have the authors made all data underlying the findings in their manuscript fully available?

Reviewer #1: Yes

Reviewer #2: Yes

4. Is the manuscript presented in an intelligible fashion and written in standard English?

Reviewer #1: Yes

Reviewer #2: Yes

5. Review Comments to the Author

Reviewer #1: Search strategy does not state the period of the studies reviewed (20xx to 20xx). Authors only state that they systematically

searched for relevant publications between March 14 and June 30, 2023. In the results you state that The final analysis included 13 distinct studies 186 from 16 articles, all published between 2010 and 2021. Was 2010-2021 the period of review or you left this open to see what you would find and this is what came up?

Line 161: There were no limitations on the date- what does this mean? Please clarify

Line 126: the health outcomes of patients with diabetes mellitus. There is no further mention or explanation of the outcomes nor are they mentioned in the screening and selection criteria.

Line 183: You mention that an article was excluded due to mismatches in the study population (n = 1)- can you explain because the study population was not clear to me. In the results you also reported on studies that included children and adults.

The description of the characteristics of the studies in the systematic review was good

Table 1, Row 6 column seven- check typo ad (and). Mean difference in HbA1c between intervention ad control group significant

On page 15 you state that 'The frequency of sessions varied from 4 to 16 (mean frequency = 8)'- do you mean of the group, individual or technology mediated sessions or all three?

Overall very strong

Reviewer #2: Thank you for the opportunity to review this study which seeks to explore the role of TPE programs in the health outcomes of individuals with diabetes in sub-Saharan Africa. The study is timely, given the limited data on the topic in the SSA region. It is well-written and interesting to read. I think it will add to and extend the existing knowledge on the topic in the literature. It is, thus, publishable.

Below are some suggestions which I think the authors may use to improve the overall quality of the paper:

1. Since the authors talk about diabetes in general, readers will benefit more if they include in the introduction a brief description of the types of diabetes: type 1, type 2, gestational diabetes, etc.

2. In lines 122 and 123, the authors state “No systematic review has been done on the evaluation of diabetes TPE programs in sub-Saharan Africa. This statement may not hold entirely. There is a similar review (Kumah E, Otchere G, Ankomah SE, Fusheini A, Kokuro C, Aduo-Adjei K, et al. Diabetes self-management education interventions in the WHO African Region: A scoping review. PLoS ONE 2021;16(8): e0256123. https://doi.org/10.1371/journal.pone.0256123) that has been conducted on the topic in the African Region. Although the authors did not specifically state that the study evaluates diabetes patient education interventions, they ended up evaluating the effectiveness of the included interventions. I would therefore suggest that the authors cite this study and state that the current study adds to and extends this existing study. And sub-Saharan Africa is not different from the WHO African region. Thus, some similarity exists between the current study and the previous study. On the other hand, if the authors see no semblance between their study and the existing one, they need to cite this existing study and indicate how it is different from the present study.

3. Search strategy: I would include at least one African health electronic database, such as African Index Medicus (AIM) in the electronic database searching since the study is focused on the SSA region.

4. Line 163, what do the authors mean by "uncontrolled studies" and how is this different from observational studies?

5. Figure 1: What do the authors mean by additional records identified through other sources? Since apart from the electronic database searching, they did not indicate additional searches done through other sources, which in my opinion could have been done by screening the reference lists of the eligible papers.

6. Results: Lines 184 and 185, further explanation is needed for excluding a mixed methods study from the review, because this may have a quantitative aspect that might meet the inclusion criteria. Sometimes, in mixed methods studies, qualitative aspects are added to explain the findings of the quantitative aspects. Thus, the excluded stud(ies) may have a diabetes education intervention, with a qualitative arm.

7. Instead of diabetic patients or patients with diabetes used throughout the study, could the authors use "individuals with diabetes" or "persons with diabetes" or "people with diabetes"

8. I feel Table 1 is overloaded. Could the authors break this down into two tables instead?

9. I would present the risk of bias assessment results before presenting the main findings of the study.

Thank you

6. PLOS authors have the option to publish the peer review history of their article (what does this mean?). If published, this will include your full peer review and any attached files.

Reviewer #1: No

Reviewer #2: **Yes: **Emmanuel Kumah

---

## [Author Response · Author response to Decision Letter 0]

26 Dec 2023

Rebuttal Letter for Journal Submission

 23 December 2023

Francis Xavier Kasujja

Academic Editor

PLOS ONE

Dear Editor,

Thank you for inviting us to submit a revised version of our manuscript entitled 'Therapeutic Patient Education Programs on Diabetes in Sub-Saharan Africa: A Systematic Review' for potential publication in PLOS ONE. We appreciate the time and effort you and the reviewers have dedicated to providing feedback on ways to strengthen our paper. We have incorporated changes based on the suggestions provided and are resubmitting our article for further consideration.

For your convenience, we have attached a document containing a point-by-point response to the questions and comments we received. We are thankful for the opportunity to refine our manuscript and look forward to hearing from you regarding our submission. If there are any further questions or comments, we are available to address them.

We trust that the revisions and our accompanying response align with the standards of PLOS ONE, making our manuscript suitable for publication. Thank you for your time and attention to this matter.

Sincerely,

Jean Toniolo

Author

Responses to Reviewers’ Comments

Journal : PLOS ONE 

Submission ID : PONE-D-23-34477

Title : Therapeutic patient education programs on diabetes in sub-Saharan Africa: A systematic review

Authors : Omomene Iwelomen, Jean Toniolo, Pierre-Marie Preux, Pascale Beloni

We express our gratitude to the reviewers for their meticulous review of our manuscript and for offering valuable comments and suggestions. We have diligently attended to all the issues highlighted in the review report, and we trust that our revisions have elevated the paper to meet an acceptable standard.

Academic Editor

The manuscript has been revised to meet PLOS ONE ‘s style requirement as well as for the file naming.

2. We note that this manuscript is a systematic review or meta-analysis; our author guidelines therefore require that you use PRISMA guidance to help improve reporting quality of this type of study. Please upload copies of the completed PRISMA checklist as Supporting Information with a file name "PRISMA checklist ». 

Our systematic review was very well conducted using the PRISMA guidance and was mentioned in the methods section of our review. A PRISMA checklist was also included as a supporting information file S2 table, we renamed the file PRISMA checklist.

Responses to Reviewers’ Comments

Reviewer #1

 Is the manuscript technically sound, and do the data support the conclusions?

Reviewer #1: Yes 

We thank the reviewer for the favorable response.

1. Search strategy does not state the period of the studies reviewed (20xx to 20xx). 

Authors only state that they systematically searched for relevant publications between March 14 and June 30, 2023. In the results you state that the final analysis included 13 distinct studies 186 from 16 articles, all published between 2010 and 2021. Was 2010-2021 the period of review or you left this open to see what you would find, and this is what came up? There was no limit to the date the articles were published to have been selected. This was decided upon in order to be more exhaustive. The reference of ‘2010 to 2021’ in our result section is the range of date of publication for the already eligible articles.

2. Line 161: There were no limitations on the date- what does this mean? Please clarify. 

Please see response to comment n°1 above. Date of publication of articles was not among our selection criteria.

3. Line 126: the health outcomes of patients with diabetes mellitus. There is no further mention or explanation of the outcomes, nor are they mentioned in the screening and selection criteria. 

We have reformulated ‘health outcomes’ as ‘health-related outcomes’ which may include glycemic control, BMI, self-management knowledge and medication adherence amongst others. We have also mentioned in our selection criteria that interventions must have measured a change in health-related outcomes.

4. Line 183: You mention that an article was excluded due to mismatches in the study population (n = 1)- can you explain because the study population was not clear to me. In the results you also reported on studies that included children and adults. 

Mismatches in the study population was in reference to studies done outside the sub-Saharan region and not based on age. We have added some clarifications in the article.

5. The description of the characteristics of the studies in the systematic review was good. 

We thank the reviewer for their kind comment.

6. Table 1, Row 6 column seven- check typo ad (and). Mean difference in HbA1c between intervention ad control group significant. 

We have made the correction on the typo.

7. On page 15 you state that 'The frequency of sessions varied from 4 to 16 (mean frequency = 8)'- do you mean of the group, individual or technology mediated sessions or all three? 

The frequency of the sessions was in reference to the group sessions. We have made the clarification in the article.

8. Overall very strong. We thank the reviewer for their kind comment.

Reviewer #2

 Thank you for the opportunity to review this study which seeks to explore the role of TPE programs in the health outcomes of individuals with diabetes in sub-Saharan Africa. The study is timely, given the limited data on the topic in the SSA region. It is well-written and interesting to read. I think it will add to and extend the existing knowledge on the topic in the literature. It is, thus, publishable. 

We appreciate the reviewer’s kind comment.

1. Since the authors talk about diabetes in general, readers will benefit more if they include in the introduction a brief description of the types of diabetes: type 1, type 2, gestational diabetes, etc. We thank the reviewer for their suggestion. 

While we talk about diabetes in general, we made it clear that our study will only include type 1 and type 2 diabetes. We believe what is more pertinent to understand is the effect of badly managed diabetes which we described in the introduction. 

We also mentioned in our introduction that type 2 diabetes has modifiable risk factors to give a hint to our readers the peculiarity of this type of diabetes.

2. In lines 122 and 123, the authors state “No systematic review has been done on the evaluation of diabetes TPE programs in sub-Saharan Africa. This statement may not hold entirely. There is a similar review (Kumah E, Otchere G, Ankomah SE, Fusheini A, Kokuro C, Aduo-Adjei K, et al. Diabetes self-management education interventions in the WHO African Region: A scoping review. PLoS ONE 2021;16(8): e0256123. https://doi.org/10.1371/journal.pone.0256123) that has been conducted on the topic in the African Region. Although the authors did not specifically state that the study evaluates diabetes patient education interventions, they ended up evaluating the effectiveness of the included interventions. I would therefore suggest that the authors cite this study and state that the current study adds to and extends this existing study. And sub-Saharan Africa is not different from the WHO African region. Thus, some similarity exists between the current study and the previous study. On the other hand, if the authors see no semblance between their study and the existing one, they need to cite this existing study and indicate how it is different from the present study. 

We thank the reviewer for bringing this to our focus. We have cited this study as a similar review. However, we have stated that while similar… ‘our study seeks to take a more systematic approach and a different methodology in view to offer our unique perspective on therapeutic patient education programs on diabetes while treating the sub-Saharan African region as the distinct area of focus.’

3. Search strategy: I would include at least one African health electronic database, such as African Index Medicus (AIM) in the electronic database searching since the study is focused on the SSA region. 

While we carefully selected the databases to the best of our knowledge that can guarantee an optimal coverage, we accept that this can be a limitation of our study and have included this information in our article.

4. Line 163, what do the authors mean by "uncontrolled studies" and how is this different from observational studies? 

We have reformulated the phrase as ‘studies with no control group’ which is what we initially intended to pass across.

5. Figure 1: What do the authors mean by additional records identified through other sources? Since apart from the electronic database searching, they did not indicate additional searches done through other sources, which in my opinion could have been done by screening the reference lists of the eligible papers. 

We have removed the column ‘additional records identified through other sources’ from Figure 1 signifying that we only searched for articles from our chosen databases which we had imagined to be exhaustive.

6. Results: Lines 184 and 185, further explanation is needed for excluding a mixed methods study from the review, because this may have a quantitative aspect that might meet the inclusion criteria. Sometimes, in mixed methods studies, qualitative aspects are added to explain the findings of the quantitative aspects. Thus, the excluded stud(ies) may have a diabetes education intervention, with a qualitative arm. 

‘Mixed method study’ was in reference to studies with compound interventions (increased physician contact time, physician training for overall management of diabetes, etc.) such that it was impossible to isolate the education intervention given. 

The studies may have also had more than one reason for ineligibility.

We have reformulated ‘mixed method’ as ‘non-related interventions’.

7. Instead of diabetic patients or patients with diabetes used throughout the study, could the authors use "individuals with diabetes" or "persons with diabetes" or "people with diabetes”. 

We thank the reviewer for their suggestion. We have effectuated the suggestion.

8. I feel Table 1 is overloaded. Could the authors break this down into two tables instead? 

We thank the reviewer for bring this to our notice. Our idea of the table was to give an overview of all important aspects of each intervention study in a coherent manner which we believe will be disrupted by breaking down the table.

We have however made the table more concise and less charged by eliminating repetitions and utilizing more symbols.

9. I would present the risk of bias assessment results before presenting the main findings of the study. 

While we agree it is a good idea to present the risk of bias assessment before the main findings, we presented our risk of bias assessment with some methodological limitations of the studies which can only be comprehensible by reading and understanding the main findings.

---

## [Decision Letter · Decision Letter 1]

13 Feb 2024

Therapeutic patient education programs on diabetes in sub-Saharan Africa: A systematic review

PONE-D-23-34477R1

Dear Dr. Toniolo,

We’re pleased to inform you that your manuscript has been judged scientifically suitable for publication and will be formally accepted for publication once it meets all outstanding technical requirements.

Kind regards,

Francis Xavier Kasujja

Academic Editor

PLOS ONE

Additional Editor Comments (optional):

Reviewers' comments:

Reviewer's Responses to Questions

**Comments to the Author**

1. If the authors have adequately addressed your comments raised in a previous round of review and you feel that this manuscript is now acceptable for publication, you may indicate that here to bypass the “Comments to the Author” section, enter your conflict of interest statement in the “Confidential to Editor” section, and submit your "Accept" recommendation.

Reviewer #2: (No Response)

2. Is the manuscript technically sound, and do the data support the conclusions?

Reviewer #2: (No Response)

3. Has the statistical analysis been performed appropriately and rigorously? 

Reviewer #2: (No Response)

4. Have the authors made all data underlying the findings in their manuscript fully available?

Reviewer #2: (No Response)

5. Is the manuscript presented in an intelligible fashion and written in standard English?

Reviewer #2: (No Response)

6. Review Comments to the Author

Reviewer #2: (No Response)

7. PLOS authors have the option to publish the peer review history of their article (what does this mean?). If published, this will include your full peer review and any attached files.

Reviewer #2: No

---

## [Editor Report · Acceptance letter]

23 Mar 2024

PONE-D-23-34477R1 

PLOS ONE

Dear Dr. Toniolo, 

I'm pleased to inform you that your manuscript has been deemed suitable for publication in PLOS ONE. Congratulations! Your manuscript is now being handed over to our production team.

Kind regards, 

on behalf of

Dr. Francis Xavier Kasujja 

Academic Editor

PLOS ONE